# Adaptive Fuzzy Event-Triggered Cooperative Control for Multi-Robot Systems: A Predefined-Time Strategy

**DOI:** 10.3390/s23187950

**Published:** 2023-09-18

**Authors:** Xuehong Tian, Xin Huang, Haitao Liu, Qingqun Mai

**Affiliations:** 1Shenzhen Institute of Guangdong Ocean University, Shenzhen 518120, China; gdtianxh@126.com (X.T.); qingqmai@163.com (Q.M.); 2School of Mechanical Engineering, Guangdong Ocean University, Zhanjiang 524088, China

**Keywords:** predefined-time cooperative control, asymmetric tan-type barrier Lyapunov function, predefined-time fuzzy logic system, dynamic relative threshold event triggering, multi-robot systems

## Abstract

A predefined-time adaptive fuzzy cooperative controller with event triggering is proposed for multi-robot systems that takes into account external disturbances, input saturation, and model uncertainties in this paper. First, based on the asymmetric tan-type barrier Lyapunov function, a predefined-time controller is proposed to acquire a quick response and more precise convergence time under the directed communication topology. Second, predefined-time fuzzy logic systems are developed to approximate external disturbances and model uncertainties. Third, a dynamic relative threshold event-triggered mechanism is improved to save the communication resources of the robots. Subsequently, the proof procedure for the predefined-time stability is given using the Lyapunov stability theorem. Finally, some simulation examples, including a comparative experi-ment on multi-robot systems, are provided to test the effectiveness of the above algorithm.

## 1. Introduction

For many research areas and practical applications, the convergence rate of the system error is a critically important index to assess the superiority of a controller, such as spacecraft [1], mobile robots [2], manipulators [3], autonomous underwater vehicles [4], and others [5,6]. For this indicator, there are various studies about deadbeat control [7], finite-time control [8], and fixed-time control [9]. In [10], for non-affine pure-feedback multi-agent systems, a finite-time control law is developed to ensure that the systems are finite-time-stable. Finite-time control has the advantages of fast convergence speed and strong robustness compared to asymptotically convergent control. However, under the action of a finite-time controller, the stabilization time is affected by the initial states of the system. The fixed-time stability theorem was developed to ensure that the stabilization time is affected by any initial state of the system. The fixed-time theorem is applied to the design of the controller and observer. A fixed-time fault-tolerant consensus tracking problem is addressed by a fixed-time fault-tolerant local control protocol in [11]. For multi-agent systems that consider disturbances, [12] investigates an event-triggered consensus control law and proves that the systems are practically fixed-time-stable. However, the current setting time estimation technologies, including finite time and fixed time, are usually inaccurate and conservative. The biggest disadvantage of those estimation technologies is that the estimated settling time is the maximum value, and the calculation of the maximum stable time will be affected by the parameters of the controller or the system. It is difficult to find suitable control parameters so that the actual settling time is the same as the estimated maximum time. Therefore, the predefined-time theorem is proposed in [13]. The predefined-time theorem is elaborated and proven, and the predefined-time first-order sliding mode controller is designed. In [14], for high-order integrator systems, a control method is proposed that achieves predefined-time convergence. In [13], a novel predefined-time stabilization formulation is developed for a class of second-order systems. The stabilization time of the controller or observer designed by using the predefined-time theory is not affected by the system initial state parameter and does not require complex parameter design, so an accurate convergence time can be obtained. At present, however, the research on predefined-time controllers is relatively small and has not formed a scale.

How to strengthen controller robustness has been a popular research topic. A large number of control technologies have been developed for multi-agent systems or nonlinear systems, for example, backstepping control [15], robust control [16], adaptive control [17], and dynamic surface control [18,19]. The robustness of the controller can be effectively improved by limiting the error. In [20], for a nonlinear aircraft system, the outputs are limited by utilizing a symmetric log-type barrier Lyapunov function (BLF). In [21], to achieve improved control performance and robustness, the full-state constrained control of the Euler-Lagrange system is achieved by utilizing the BLF technique to limit the error within a preset range. Compared with symmetric BLF, asymmetric BLF in [22] has better error-limiting effects.

In practice, external disturbances have a significant impact on the control performance. Disturbance observers, neural networks, and other compensation methods have been developed to address this problem. In [23,24,25], the estimated value of disturbances is obtained by utilizing the disturbance observer. The extended state observer in [26,27] can observe not only the external disturbances and uncertainties but also the state of the system. Neural network techniques such as radial basis function neural networks (RBFNNs) [28] are widely applied to deal with dynamic nonlinearities in systems. A multilayer perceptron neural network [29] is deployed to achieve a consensus protocol. Nevertheless, the parameters of the observer are complex to determine, and the computational burden of the RBFNN is enormous. To make it easier to apply in practice, a fuzzy logic system was developed for a multi-agent system in [30]. Since it is rule-based, the fuzzy control mechanism is easy to understand, design, and implement. In [31], a fuzzy control is used and proves asymptotic convergence of the system. A consensus control strategy based on the fixed-time fuzzy logic system has evolved to ensure that all consensus errors converge to the origin in a fixed time in [32]. In [33], resilient fuzzy stabilization is developed for discrete-time Takagi–Sugeno systems, which is capable of providing much less conservative results than conventional fuzzy stabilization. The system has stronger robustness and faster convergence than the asymptotically convergent fuzzy logic system. However, it is still not a simple task to obtain the exact convergence time of the fixed-time fuzzy logic system.

In most cases, the communication bandwidth between actuators is restricted. It becomes a challenge to design a controller that saves communication bandwidth resources. Designing different event-triggering strategies [34] has become an effective way to conserve resources. For the multi-agent systems, the leader’s states are acquired by a distributed observer, which is controlled by an event-triggered controller in [16]. In [35], an event-triggered strategy is used to save communication among agents while ensuring synchronization accuracy. For the cluster synchronization control of coupled neural networks, a new data sampling and triggering mechanism is designed to reduce the communication burden in [36]. An adaptive event-based interval type-2 (IT-2) fuzzy security control is proposed for networked control systems with multiple network attacks [37]. In [38], to achieve formation control, some event-triggered mechanisms were developed, including static and dynamic event-triggered mechanisms.

Motivated by the status of the above study, a predefined-time adaptive fuzzy controller with a novel event-triggered mechanism is developed to ensure that the whole closed-loop system is predefined-time-stable for multi-robot systems while considering the external disturbances and model uncertainties. The contributions are mainly threefold, as listed below.

(1) A predefined-time cooperative controller based on an asymmetric tan-type BLF is designed. The aim is to ensure that position tracking error is guaranteed in a certain range and that all errors of the system are predefined-time-stable for multi-robot systems. Different from the BLF [22], the controller is capable of greater robustness and more accurate determination of the convergence time of the system.

(2) Different from the other methods [15,26], a fuzzy logic system is used to approximate the external disturbances and model uncertainties to improve the accuracy. Moreover, it has been shown that fuzzy logic systems are predefined-time-stable, which is the greatest advantage.

(3) A dynamic event-triggered mechanism is proposed based on a relative threshold strategy to effectively conserve the communication bandwidth and communication. The strategy can ensure that the controller has a good control effect regardless of the degree change in the size of the control input. Different from [39], the proposed dynamic relative threshold event-triggered controller is designed with a different theorem for better performance. A dynamic function is designed to dynamically adjust the inter-execution intervals and provide a guarantee for effective operation of the system.

The rest of the article is generalized as follows. Preliminary knowledge is presented in Section 2, including definitions, model descriptions, and concepts. In Section 3, the predefined-time adaptive fuzzy cooperative controller is proposed. Simulations and conclusions are presented in Section 4 and Section 5, respectively.

## 2. Preliminaries and Problem Formulation

### 2.1. Model Description

Consider multi-robot systems as follows.
(1)Mi(qi)q¨i+Ci(qi,q˙i)q˙i+Gi(qi)=ui+ρoi
where Ci(qi,q˙i)∈ℝn×n denotes the Coriolis and centrifugal forces. The symmetric positive definite inertia matrix is denoted as Mi(qi)∈ℝn×n. qi∈ℝn and q˙i∈ℝn are the position and velocity of the follower robot, respectively. Gi(qi)∈ℝn denotes the gravitational force. ui∈ℝn is the control input of the i-th follower robot without input saturation, and ρoi∈ℝn represents the uncertainties. Input saturation is defined as follows:(2)uci={ui+,ifui>ui+ui,ifui−≤ui≤ui+ui−,ifui<τι−
where uci is the actual control input of the i-th follower robot with input saturation and ui+ and ui− are the upper and lower bounds of input saturation, respectively.

### 2.2. Graph Theory Description

Denote the follower set and leader set by F={1,…,N} and L={1,…,M}, respectively. A directed graph G=(V,E,A) is used to describe the condition of the information of status exchanges among agents, where $A=[a_{ij}]\in R^{N\times N}$ indicates the communication of followers. $a_{ij}=1,j\neq i$ means that the i-th follower can receive information from the j-th follower; otherwise, $a_{ij}=0$.
$V=\left\{v_{1},v_{2},\ldots ,v_{N}\right\}$ is the set of followers. $D=\mathrm{diag}\left\{d_{i}\right\},i=1,\ldots ,N$ and $d_{i}=\sum_{j=1}^{N}a_{ij}$. L'=D−A is the Laplacian matrix. The adjacency matrix of leaders $B=[b_{ir}]\in R^{N\times M}$ indicates the communication among the leaders and followers. If the i-th follower can receive information from the r-th leader, $b_{ir}=1$; otherwise, $b_{ir}=0$.

### 2.3. Fuzzy Logic Systems Description

Fuzzy logic systems (FLSs) are composed of fuzzifiers, defuzzifiers, fuzzy engines, and fuzzy IF-THEN rules. The vectors x=[x1,x2,…,xn]T∈ℝn and f^∈ℝ denote the input and output of the fuzzy inference engine, respectively. In the r-th fuzzy rule, ℝ(r), Air,i∈n and Br represent the fuzzy set and the output of the fuzzy singleton, respectively. If xi is Air, then f^ is Br.

The technologies are applied in the FLSs, including the center-average defuzzifier singleton fuzzifier and product inference, and $\hat{f}\left(x\right)$ is designed as
(3)$$\hat{f}\left(x\right)=\frac{\sum_{i=1}^{N}B_{r}\left[\prod_{i=1}^{n}\mu_{A_{i}^{r}}\left(x_{i}\right)\right]}{\sum_{i=1}^{N}\left[\prod_{i=1}^{n}\mu_{A_{i}^{r}}\left(x_{i}\right)\right]}={\hat{\theta }}^{T}\psi \left(x\right)$$
where $\mu_{A_{i}^{r}}\left(x_{i}\right)$ is the membership degree of $x_{i}$ to $A_{i}^{r}$, N is the total number of fuzzy rules, the adjustable parameter vector is denoted as $\hat{\theta }={\left[{\hat{\theta }}_{1},{\hat{\theta }}_{2},\ldots ,{\hat{\theta }}_{N}\right]}^{T}$, and $\psi \left(x\right)={\left[\psi_{1}\left(x\right),\psi_{2}\left(x\right),\ldots ,\psi_{N}\left(x\right)\right]}^{T}$ is a fuzzy basis vector, where
(4)$$\psi_{r}\left(x\right)=\frac{\prod_{i=1}^{n}\mu_{A_{i}^{r}}\left(x_{i}\right)}{\sum_{r=1}^{N}\left[\prod_{i=1}^{n}\mu_{A_{i}^{r}}\left(x_{i}\right)\right]}$$

To ensure that there is at least one active rule, the following assumption is given: The fuzzy basis functions are chosen, i.e., $\sum_{r=1}^{N}\left[\prod_{i=1}^{n}\mu_{A_{i}^{r}}\left(x_{i}\right)\right]>0$.

For analysis purposes, θ is regarded as the ideal parameter of $\hat{\theta }$, which is designed as
(5)$$\theta =\arg\underset{\hat{\theta }}{\min}\left[\underset{t}{\sup}\left|f\left(x\right)-\hat{f}\left(x\right)\right|\right]$$

Assume that the uncertainty $f_{i}$ is represented as
(6)$$f_{i}\left(x\right)=\theta_{i}^{T}\psi_{i}\left(x\right)+\varepsilon_{i}\left(x\right),\left|\varepsilon_{i}\left(x\right)\right|\leq {\bar{\lambda }}_{i}$$
where ${\bar{\lambda }}_{i}>0$ is an unknown constant and ε(x) is the error of approximation.

### 2.4. Notions

For stability analysis purposes, Θi is defined as Θi=‖θi‖2. |∗| represents the absolute value. argmin{∗} stands for minimized {∗}. ⎡∗⎦q stands for |∗|qsign(∗). ‖∗‖ represents the Euclidean norm; ∗^ and ∗˜ represent the approximation and approximation error for ∗, respectively. The real number set is marked as ℝ, where ℝN×N and ℝN denote the N×N-dimensional and N-dimensional Euclidean spaces, respectively.

### 2.5. Assumption and Lemmas

 **Assumption 1.** 
*The communication topology G is directed. For each follower i∈F, there exists at least one leader r∈L that has a directed path to that follower.*


 **Assumption 2.** *Suppose that there exists a known constant Θi0>0 such that Θi≤Θi0, where Θi∈ℝ*.

**Lemma** **1 [40].***For any constant ∂>0 and ς∈ℝ, one has $0\leq \left|\varsigma \right|-\frac{\varsigma^{2}}{\sqrt{\varsigma^{2}+\partial^{2}}}\leq \partial$*.

**Lemma** **2 [41].***System $\dot{\chi }=g\left(\chi \right),\chi \left(0\right)=\chi_{0}$ is practically a predefined time table, and the stable time is $2T_{c}$ if there exists a Lyapunov function V satisfying the following condition: $\dot{V}\leq -\frac{\pi }{rT_{c}}\left(V^{1-\frac{r}{2}}+V^{1+\frac{r}{2}}\right)+b$, 0<r<1, $T_{c}>0$ and b>0*.

**Lemma** **3 [42].***For any constant satisfying p>0,z≤p, if h>1, one has ${\left(z-p\right)}^{h}\leq z^{h}-p^{h}$; if h>0, one has $p^{h}\left(z-p\right)\leq \frac{1}{1+h}\left(z^{1+h}-p^{1+h}\right)$*.

## 3. Main Results

In this section, a predefined-time adaptive fuzzy controller based on BLF is introduced for multi-robot systems (MRSs). An asymmetric tan-type BLF is considered to solve error time-varying constraint problems, and fuzzy logic systems (FLSs) are used to solve the interference problem of complicated disturbances and unknown nonlinearity. Additionally, an event-triggered mechanism is implemented to decrease robot communication. Finally, some mathematical proof procedures are given to demonstrate that the multi-robot systems are predefined-time-stable.

### 3.1. Predefined-Time Control Based on BLF

In this part, without considering external disturbances and system uncertainties, a predefined-time controller based on BLF is developed under the directed topology for multi-robot systems, which proves that the systems are predefined-time-stable.

For the sake of design, let $x_{i}=q_{i}$ and $v_{i}={\dot{q}}_{i}$. Thus, the i-th follower is expressed as
(7)$$\left\{ \begin{array}{l} {\dot{x}}_{i}=v_{i},\\ {\dot{v}}_{i}=M_{i}{}^{-1}\left(u_{i}+\rho_{oi}-G_{i}-C_{i}v_{i}\right). \end{array}\right.$$

The r-th leader is expressed as
(8)$$\left\{ \begin{array}{l} {\dot{x}}_{r}=v_{r},\\ {\dot{v}}_{r}=u_{r}. \end{array}\right.$$

The position error of the i-th follower is as follows:(9)$$e_{xi}(t)=\sum_{j=1}^{N}a_{ij}(x_{i}(t)-x_{j}(t))+\sum_{r=1}^{M}b_{ir}(x_{i}(t)-x_{r}(t))$$

The differentiation of Equation (9) is as follows:(10)$$e_{vi}(t)=\sum_{j=1}^{N}a_{ij}(v_{i}(t)-v_{j}(t))+\sum_{r=1}^{M}b_{ir}(v_{i}(t)-v_{r}(t))$$

The predefined-time controller design process contains two steps.

Step 1: Observed from (9) and to stabilize the error $e_{xi}$, the virtual control law $\alpha_{i}$ of the i-th follower is devised.

The asymmetric tan-type BLF is chosen, and the specific form is as follows:(11)$$V_{1}=\left(1-q\left(e_{xi}\right)\right)\frac{L_{Li}{}^{2}}{\pi }\tan\left(\frac{\pi e_{xi}{}^{2}}{2L_{Li}{}^{2}}\right)+q\left(e_{xi}\right)\frac{L_{Ui}{}^{2}}{\pi }\tan\left(\frac{\pi e_{xi}{}^{2}}{2L_{Ui}{}^{2}}\right)$$
(12)$$q(e_{xi})=\left\{ \begin{array}{l} 1,\;e_{xi}>0\\ 0,\;e_{xi}\leq 0 \end{array}\right.$$
where $L_{Li}$ and $L_{Ui}$ are the preset boundaries. $L_{Li}$ and $L_{Ui}$ are functions of time, and $L_{Li}$ > 0, $L_{Ui}$ > 0.

Then, the derivative of $V_{1}$ yields
(13)$${\dot{V}}_{1}=\left(1-q\left(e_{xi}\right)\right)\left[\frac{2L_{Li}{\dot{L}}_{Li}}{\pi }\tan\left(\frac{\pi e_{xi}{}^{2}}{2L_{Li}{}^{2}}\right)+\Lambda_{Li}{\dot{e}}_{xi}-\Lambda_{Li}e_{xi}\frac{{\dot{L}}_{Li}}{L_{Li}}\right]+q\left(e_{xi}\right)\left[\frac{2L_{Ui}{\dot{L}}_{Ui}}{\pi }\tan\left(\frac{\pi e_{xi}{}^{2}}{2L_{Ui}{}^{2}}\right)+\Lambda_{Ui}{\dot{e}}_{xi}-\Lambda_{Ui}e_{xi}\frac{{\dot{L}}_{Ui}}{L_{Ui}}\right]$$
where $\Lambda_{Li}=\frac{e_{xi}}{\cos^{2}\left(\frac{\pi e_{xi}{}^{2}}{2L_{Li}{}^{2}}\right)},\Lambda_{Ui}=\frac{e_{xi}}{\cos^{2}\left(\frac{\pi e_{xi}{}^{2}}{2L_{Ui}{}^{2}}\right)}$ and the initial state satisfies $-L_{Li}(0)<e_{xi}(0)<L_{Ui}(0)$.

 **Remark** **1.** 
*Observed from (11), one has*

(14)
$$\left\{ \begin{array}{l} \underset{e_{xi}\to 0^{+}}{\lim}V_{1}=\underset{e_{xi}\to 0^{-}}{\lim}V_{1}=0\\ \underset{e_{xi}\to L_{Li}}{\lim}V_{1}=\underset{e_{xi}\to L_{Ui}}{\lim}V_{1}=\infty \end{array}\right.$$

* where the state $e_{xi}$ follows $-L_{Li}\left(t\right)<e_{xi}\left(t\right)<L_{Ui}\left(t\right)$ if the initial value satisfies $-L_{Li}\left(0\right)<e_{xi}\left(0\right)<L_{Ui}\left(0\right)$. When system states are unconstrained, such as $L_{Li}\to \infty$ and $L_{Ui}\to \infty$, using L’ Hospital theory:*

(15)
$$\underset{L_{Li}\to \infty ,L_{Ui}\to \infty }{\lim}V_{1}=\frac{1}{2}e_{xi}{}^{2}$$


*Thus, the i-th virtual controller $\alpha_{i}$ is developed as*

(16)
$$\begin{array}{ll} \alpha_{i} & =-\frac{\Xi_{Li}\left(\left(1-q\left(e_{xi}\right)\right)L_{Li}{}^{2}\sin\left(\frac{\pi e_{xi}{}^{2}}{L_{Li}{}^{2}}\right)+q\left(e_{xi}\right)L_{Ui}{}^{2}\sin\left(\frac{\pi e_{xi}{}^{2}}{L_{Ui}{}^{2}}\right)\right)}{\pi e_{xid}}+\Xi_{Li}e_{xi}+{\left(\sum_{j=1}^{N}a_{ij}+\sum_{r=1}^{M}b_{ir}\right)}^{-1}\left(\sum_{j=1}^{N}a_{ij}v_{j}+\sum_{r=1}^{M}b_{ir}v_{r}\right)\\ & -\Xi_{\Lambda i}{}^{-1}\frac{\pi }{rT_{c}}((\left(1-q\left(e_{xi}\right)\right)\frac{L_{Li}{}^{2}}{\pi }\tan\left(\frac{\pi e_{xi}{}^{2}}{2L_{Li}{}^{2}}\right){+q\left(e_{xi}\right)\frac{L_{Ui}{}^{2}}{\pi }\tan\left(\frac{\pi e_{xi}{}^{2}}{2L_{Ui}{}^{2}}\right))}^{1-\frac{r}{2}}+(\left(1-q\left(e_{xi}\right)\right)\frac{L_{Li}{}^{2}}{\pi }\tan\left(\frac{\pi e_{xi}{}^{2}}{2L_{Li}{}^{2}}\right)\\ & +{q\left(e_{xi}\right)\frac{L_{Ui}{}^{2}}{\pi }\tan\left(\frac{\pi e_{xi}{}^{2}}{2L_{Ui}{}^{2}}\right))}^{1+\frac{r}{2}}) \end{array}$$

* where 0<r<1, $T_{c}$ is a positive constant, $\Xi_{\Lambda i}=\left(\left(1-q\left(e_{xi}\right)\right)\Lambda_{Li}+q\left(e_{xi}\right)\Lambda_{Ui}\right)\left(\sum_{j=1}^{N}a_{ij}+\sum_{r=1}^{M}b_{ir}\right)$ and $\Xi_{Li}=\left(\left(1-q\left(e_{xi}\right)\right)\frac{{\dot{L}}_{Li}}{L_{Li}}+q\left(e_{xi}\right)\frac{{\dot{L}}_{Ui}}{L_{Ui}}\right){\left(\sum_{j=1}^{N}a_{ij}+\sum_{r=1}^{M}b_{ir}\right)}^{-1}$
*


**Step 2:** Observed from $e_{\alpha i}=v_{i}-\alpha_{i}$ and to stabilize the error $e_{\alpha i}$, the i-th actual controller $u_{i}$ is devised as
(17)$$u_{i}=G_{i}+C_{i}v_{i}-\rho_{io}+M_{i}\left(-\frac{\pi }{rT_{c}}\left(\frac{1}{2^{1-\frac{r}{2}}}e_{\alpha i}{}^{1-r}+\frac{1}{2^{1+\frac{r}{2}}}e_{\alpha i}{}^{1+r}\right)+{\dot{\alpha }}_{i}-\Xi_{\Lambda i}\right)$$
where $0<r<1,T_{c}>0$.

**Theorem** **1.**
*For a class of MRSs (7) and (8) with Assumption 1, the problem of cooperative control is realized under the predefined-time controller (16), (17). The following properties are reasonable. (a) The position errors are limited to a predetermined range. (b) The errors of consensus converge to near zero in predefined time $2T_{c}$.*


 **Proof.** The following Lyapunov function is selected:(18)$$V_{2}=V_{1}+\frac{1}{2}e_{\alpha i}{}^{2}$$Taking the derivative with respect to $V_{2}$ gives
(19)$$\begin{array}{ll} {\dot{V}}_{2} & =\Xi_{\Lambda i}\left(e_{\alpha i}+\alpha_{i}\right)+e_{\alpha i}(M_{i}^{-1}(u_{i}+\rho_{oi}-G_{i}-C_{i}v_{i})-{\dot{\alpha }}_{i})+\left(1-q\left(e_{xi}\right)\right)[\frac{2L_{Li}{\dot{L}}_{Li}}{\pi }\tan\left(\frac{\pi e_{xi}{}^{2}}{2L_{Li}{}^{2}}\right)\\ & -\Lambda_{Li}\left(\left(\sum_{j=1}^{N}a_{ij}v_{j}+\sum_{r=1}^{M}b_{ir}v_{r}\right)+e_{xi}\frac{{\dot{L}}_{Li}}{L_{Li}}\right)]+q\left(e_{xi}\right)[\frac{2L_{Ui}{\dot{L}}_{Ui}}{\pi }\tan\left(\frac{\pi e_{xi}{}^{2}}{2L_{Ui}{}^{2}}\right)\\ & -\Lambda_{Ui}\left(\left(\sum_{j=1}^{N}a_{ij}v_{j}+\sum_{r=1}^{M}b_{ir}v_{r}\right)+e_{xi}\frac{{\dot{L}}_{Ui}}{L_{Ui}}\right)] \end{array}$$Substituting (16) and (17) into (19), one has
(20)$${\dot{V}}_{2}\leq -\frac{\pi }{rT_{c}}\left(V_{1}{}^{1-\frac{r}{2}}+V_{1}{}^{1+\frac{r}{2}}+{\left(\frac{1}{2}e_{\alpha i}{}^{2}\right)}^{1-\frac{r}{2}}+{\left(\frac{1}{2}e_{\alpha i}{}^{2}\right)}^{1+\frac{r}{2}}\right)$$The proof is performed. □

### 3.2. Predefined-Time Fuzzy Logic System Design

In this part, on the basis of the above controller, fuzzy logic systems are introduced to approximate the uncertainties, and it is proven that the systems are predefined-time-stable.

Considering the external disturbances and nonlinear uncertainties, (7) is presented as
(21)$$\left\{ \begin{array}{l} {\dot{x}}_{i}=v_{i},\\ {\dot{v}}_{i}=M_{i}^{-1}u_{i}(t)+f_{i}, \end{array}\right.$$
where $f_{i}$ denotes the set of external disturbances and uncertainties, which is denoted as
(22)$$f_{i}=M_{i}^{-1}(\rho_{oi}-G_{i}-C_{i}v_{i})$$

Define the error ${\tilde{\Theta }}_{i}$ as ${\tilde{\Theta }}_{i}=\Theta_{i}-{\hat{\Theta }}_{i}$. Then, the predefined-time fuzzy control can be designed as
(23)$$u_{fi}=M_{i}(-\frac{\pi }{rT_{c}}\left(\frac{1}{2^{1-\frac{r}{2}}}{⎡e_{\alpha i}⎦}^{1-r}+\frac{1}{2^{1+\frac{r}{2}}}{⎡e_{\alpha i}⎦}^{1+r}\right)+{\dot{\alpha }}_{i}-\Xi_{\Lambda i}-\frac{1}{2a_{i}{}^{2}}{\hat{\Theta }}_{i}\psi {\left(x_{i}\right)}^{T}\psi \left(x_{i}\right)-\frac{e_{\alpha i}}{2})$$

The adaptive law is designed as follows:(24)Θi^˙=σi2ai2eαiψ(xi)Tψ(xi)−πrTc(2−r21−r2Θ^i1−r+2+r21+r2Θ^i1+r)
where $0<r<1,T_{c}>0$.

**Theorem** **2.**
*For a class of MRSs (7) and (8) considering the external disturbances and nonlinear uncertainties with Assumption 1, the problem of cooperative control is realized under the predefined-time adaptive fuzzy controller (16), (23), (24). The following properties are reasonable. (a) The position errors are limited to a predetermined range. (b) The errors of consensus converge to near zero in predefined time
$2T_{c}$.*


 **Proof** Due to the introduction of fuzzy logic systems, ${\dot{V}}_{2}$ is re-expressed as
(25)$$\begin{array}{ll} {\dot{V}}_{2} & =\Xi_{\Lambda i}\left(e_{\alpha i}+\alpha_{i}\right)+\left(1-q\left(e_{xi}\right)\right)[\frac{2L_{Li}{\dot{L}}_{Li}}{\pi }\tan\left(\frac{\pi e_{xi}{}^{2}}{2L_{Li}{}^{2}}\right)-\Lambda_{Li}\left(\left(\sum_{j=1}^{N}a_{ij}v_{j}+\sum_{r=1}^{M}b_{ir}v_{r}\right)+e_{xi}\frac{{\dot{L}}_{Li}}{L_{Li}}\right)]\\ & +q\left(e_{xi}\right)[\frac{2L_{Ui}{\dot{L}}_{Ui}}{\pi }\tan\left(\frac{\pi e_{xi}{}^{2}}{2L_{Ui}{}^{2}}\right)-\Lambda_{Ui}\left(\left(\sum_{j=1}^{N}a_{ij}v_{j}+\sum_{r=1}^{M}b_{ir}v_{r}\right)+e_{xi}\frac{{\dot{L}}_{Ui}}{L_{Ui}}\right)]+e_{\alpha i}\left(M_{i}^{-1}u_{i}-{\dot{\alpha }}_{i}+\theta_{i}{}^{T}\psi \left(x_{i}\right)+\varepsilon_{i}\left(x_{i}\right)\right) \end{array}$$Since $\left|\varepsilon_{i}\left(x\right)\right|\leq {\bar{\lambda }}_{i}$,
(26)$$\begin{array}{ll} {\dot{V}}_{2} & \leq \Xi_{\Lambda id}\left(e_{\alpha i}+\alpha_{i}\right)+\left(1-q\left(e_{xi}\right)\right)[\frac{2L_{Li}{\dot{L}}_{Li}}{\pi }\tan\left(\frac{\pi e_{xi}{}^{2}}{2L_{Li}{}^{2}}\right)-\Lambda_{Li}\left(\left(\sum_{j=1}^{N}a_{ij}v_{j}+\sum_{r=1}^{M}b_{ir}v_{r}\right)+e_{xi}\frac{{\dot{L}}_{Li}}{L_{Li}}\right)]\\ & +q\left(e_{xi}\right)[\frac{2L_{Ui}{\dot{L}}_{Ui}}{\pi }\tan\left(\frac{\pi e_{xi}{}^{2}}{2L_{Ui}{}^{2}}\right)-\Lambda_{Ui}\left(\left(\sum_{j=1}^{N}a_{ij}v_{j}+\sum_{r=1}^{M}b_{ir}v_{r}\right)+e_{xi}\frac{{\dot{L}}_{Ui}}{L_{Ui}}\right)]\\ & +e_{\alpha i}\left(M_{i}^{-1}u_{fi}-{\dot{\alpha }}_{i}\right)+\left|e_{\alpha i}\right|\left\|\theta_{i}\right\|\left\|\psi \left(x_{i}\right)\right\|+\left|e_{\alpha i}\right|{\bar{\lambda }}_{i} \end{array}$$Based on **Young’s inequality**, we obtain
(27)$$\begin{array}{ll} {\dot{V}}_{2} & \leq \Xi_{\Lambda i}\left(e_{\alpha i}+\alpha_{i}\right)+\left(1-q\left(e_{xi}\right)\right)[\frac{2L_{Li}{\dot{L}}_{Li}}{\pi }\tan\left(\frac{\pi e_{xi}{}^{2}}{2L_{Li}{}^{2}}\right)-\Lambda_{Li}\left(\left(\sum_{j=1}^{N}a_{ij}v_{j}+\sum_{r=1}^{M}b_{ir}v_{r}\right)+e_{xi}\frac{{\dot{L}}_{Li}}{L_{Li}}\right)]\\ & +q\left(e_{xi}\right)[\frac{2L_{Ui}{\dot{L}}_{Ui}}{\pi }\tan\left(\frac{\pi e_{xi}{}^{2}}{2L_{Ui}{}^{2}}\right)-\Lambda_{Ui}\left(\left(\sum_{j=1}^{N}a_{ij}v_{j}+\sum_{r=1}^{M}b_{ir}v_{r}\right)+e_{xi}\frac{{\dot{L}}_{Ui}}{L_{Ui}}\right)]\\ & +e_{\alpha i}{}^{T}\left(M_{i}^{-1}u_{fi}-{\dot{\alpha }}_{i}\right)+\frac{1}{2a_{i}{}^{2}}e_{\alpha i}{}^{2}\Theta_{i}\psi {\left(x_{i}\right)}^{T}\psi \left(x_{i}\right)+\frac{1}{2}a_{i}{}^{2}+\frac{1}{2}e_{\alpha i}{}^{2}+\frac{1}{2}{\bar{\lambda }}_{i}{}^{2} \end{array}$$The following Lyapunov candidate function is considered:(28)$$V_{3}=V_{2}+\frac{1}{2\sigma_{i}}{\tilde{\Theta }}_{i}{}^{2}$$Taking the derivative with respect to $V_{3}$ gives
(29)$$\begin{array}{ll} {\dot{V}}_{3} & \leq \Xi_{\Lambda i}\left(e_{\alpha i}+\alpha_{i}\right)+\left(1-q\left(e_{xi}\right)\right)[\frac{2L_{Li}{\dot{L}}_{Li}}{\pi }\tan\left(\frac{\pi e_{xi}{}^{2}}{2L_{Li}{}^{2}}\right)-\Lambda_{Li}\left(\left(\sum_{j=1}^{N}a_{ij}v_{j}+\sum_{r=1}^{M}b_{ir}v_{r}\right)+e_{xi}\frac{{\dot{L}}_{Li}}{L_{Li}}\right)]\\ & +q\left(e_{xi}\right)[\frac{2L_{Ui}{\dot{L}}_{Ui}}{\pi }\tan\left(\frac{\pi e_{xi}{}^{2}}{2L_{Ui}{}^{2}}\right)-\Lambda_{Ui}\left(\left(\sum_{j=1}^{N}a_{ij}v_{j}+\sum_{r=1}^{M}b_{ir}v_{r}\right)+e_{xi}\frac{{\dot{L}}_{Ui}}{L_{Ui}}\right)]\\ & +e_{\alpha i}\left(M_{i}^{-1}u_{fi}-{\dot{\alpha }}_{i}\right)+\frac{1}{2a_{i}{}^{2}}e_{\alpha i}{}^{2}\Theta_{i}\psi {\left(x_{i}\right)}^{T}\psi \left(x_{i}\right)+\frac{1}{2}a_{i}{}^{2}+\frac{1}{2}e_{\alpha i}{}^{2}+\frac{1}{2}{\bar{\lambda }}_{i}{}^{2}-\frac{1}{\sigma_{i}}{\tilde{\Theta }}_{i}{\dot{\hat{\Theta }}}_{i} \end{array}$$Substituting (16) and (23) into (29), one has
(30)$${\dot{V}}_{3}\leq -\frac{\pi }{rT_{c}}\left(V_{2}{}^{1-\frac{r}{2}}+V_{2}{}^{1+\frac{r}{2}}\right)+\frac{1}{2a_{i}{}^{2}}e_{\alpha i}{}^{2}{\tilde{\Theta }}_{i}\psi {\left(x_{i}\right)}^{T}\psi \left(x_{i}\right)+\frac{1}{2}a_{i}{}^{2}-\frac{1}{\sigma_{i}}{\tilde{\Theta }}_{i}{\dot{\hat{\Theta }}}_{i}+\frac{1}{2}{\bar{\lambda }}_{i}{}^{2}$$Substituting (24) into (30), we obtain
(31)$${\dot{V}}_{3}\leq -\frac{\pi }{rT_{c}}\left(V_{2}{}^{1-\frac{r}{2}}+V_{2}{}^{1+\frac{r}{2}}\right)+\frac{1}{2}a_{i}{}^{2}+\frac{1}{2}{\bar{\lambda }}_{i}{}^{2}+\frac{\pi }{rT_{c}}\left(\frac{2-r}{2^{1-\frac{r}{2}}}{\tilde{\Theta }}_{i}{\hat{\Theta }}_{i}{}^{1-r}+\frac{2+r}{2^{1+\frac{r}{2}}}{\tilde{\Theta }}_{i}{\hat{\Theta }}_{i}{}^{1+r}\right)$$Based on **Lemma 3**, one has
(32)$$\left\{ \begin{array}{l} {\tilde{\Theta }}_{i}{\hat{\Theta }}_{i}{}^{1+r}\leq \frac{1}{2+r}\left(2{\left(\Theta_{i}\right)}^{2+r}-{\left({\tilde{\Theta }}_{i}\right)}^{2+r}\right)\\ {\tilde{\Theta }}_{i}{\hat{\Theta }}_{i}{}^{1-r}\leq \frac{1}{2-r}\left(2{\left(\Theta_{i}\right)}^{2-r}-{\left({\tilde{\Theta }}_{i}\right)}^{2-r}\right) \end{array}\right.$$Then, according to **Assumption 2**, one has
(33)$${\dot{V}}_{3}\leq -\frac{\pi }{rT_{c}}\left(V_{3}{}^{1-\frac{r}{2}}+V_{3}{}^{1+\frac{r}{2}}\right)+B_{i}$$
where $B_{i}=\frac{1}{2}a_{i}{}^{2}+\frac{1}{2}{\bar{\lambda }}_{i}{}^{2}+\frac{\pi }{rT_{c}}\left(\frac{1}{2^{-\frac{r}{2}}}\Theta_{i}{}^{2-r}+\frac{1}{2^{\frac{r}{2}}}\Theta_{i}{}^{2+r}\right)$.The proof is performed. □

**Remark** **2.**
*For the MRSs (7) and (8) under the control of (16) and (23), there exists a Lyapunov function
$V_{3}$ satisfying
${\dot{V}}_{3}\leq -\frac{\pi }{rT_{c}}\left(V_{3}{}^{1-\frac{r}{2}}+V_{3}{}^{1+\frac{r}{2}}\right)+B_{i}$ and
$B_{i}>0$; therefore, the system is predefined-time-stable, and the stable time is
$2T_{c}$.*


**Remark** **3.**
*The predefined-time adaptive fuzzy controller has more robustness and faster convergence time than the controller with asymptotic convergence [31]. The proposed controller can obtain the stable time more accurately than the controller with finite/fixed-time convergence [32].*


### 3.3. Event-Triggered Mechanism Design

To reduce communication, a time-varying relative threshold event-triggered mechanism is designed. The mechanism is designed as
(34)wi(t)=−(1+δ)(eαiufi2eαi2ufi2+∂2+eαim¯2eαi2m¯2+∂2)
(35)u¯i(t)=wi(tk),∀t∈[tk,tk+1)
(36)tk+1=inf{t∈ℝ||ei(t)|≥δ|u¯i(t)|+m}
where 0<δ<1,m>0,m¯>m1−δ and ei(t)=wi(t)−u¯i(t) denotes the measurement error.

**Theorem** **3.**
*For MRSs (7) and (8) under **Assumption 1**, replacing controller (23) with event-triggered controllers (34)–(36), all the properties in **Theorem 2** still hold.*


**Remark** **4.**
*Different from the event-triggered controller in [39], the proposed event-triggered controller is derived from **Lemma 1** and provides an alternative solution for event-triggered control of relative thresholds. A comparative experiment is given in the simulation to show that the proposed controller has better performance.*


**Remark** **5.**
*In general, the starting torque of the system is larger, and when the system is stable, the torque is smaller than the starting torque. The proposed triggering mechanism is able to obtain a long trigger interval with a large control input magnitude. After the system is stabilized, the control input magnitude decreases, and at the same time, the short trigger interval can ensure that the controller has a good control effect.*


 **Proof.** From (36), one has $w_{i}\left(t\right)=(1+\lambda_{1}(t)\delta ){\bar{u}}_{i}\left(t\right)+\lambda_{2}\left(t\right)m$ in the interval $\left[t_{k},t_{k+1}\right)$, where the time-varying parameters $\lambda_{1}\left(t\right)$ and $\lambda_{2}\left(t\right)$ satisfy $\left|\lambda_{1}\left(t\right)\right|\leq 1$ and $\left|\lambda_{2}\left(t\right)\right|\leq 1$. Thus, one has ${\bar{u}}_{i}\left(t\right)=\frac{w_{i}\left(t\right)-\lambda_{2}\left(t\right)m}{1+\lambda_{1}\left(t\right)\delta }$.Then, one has
(37)$$\begin{array}{ll} {\dot{V}}_{3} & \leq \Xi_{\Lambda i}\left(e_{\alpha i}+\alpha_{i}\right)+\left(1-q\left(e_{xi}\right)\right)[\frac{2L_{Li}{\dot{L}}_{Li}}{\pi }\tan\left(\frac{\pi e_{xi}{}^{2}}{2L_{Li}{}^{2}}\right)-\Lambda_{Li}\left(\left(\sum_{j=1}^{N}a_{ij}v_{j}+\sum_{r=1}^{M}b_{ir}v_{r}\right)+e_{xi}\frac{{\dot{L}}_{Li}}{L_{Li}}\right)]\\ & +q\left(e_{xi}\right)[\frac{2L_{Ui}{\dot{L}}_{Ui}}{\pi }\tan\left(\frac{\pi e_{xi}{}^{2}}{2L_{Ui}{}^{2}}\right)-\Lambda_{Ui}\left(\left(\sum_{j=1}^{N}a_{ij}v_{j}+\sum_{r=1}^{M}b_{ir}v_{r}\right)+e_{xi}\frac{{\dot{L}}_{Ui}}{L_{Ui}}\right)]\\ & +e_{\alpha i}\left(M_{i}^{-1}\left(\frac{w_{i}\left(t\right)-\lambda_{2}\left(t\right)m}{1+\lambda_{1}\left(t\right)\delta }\right)-{\dot{\alpha }}_{i}\right)+\frac{1}{2}a_{i}{}^{2}+\frac{1}{2}e_{\alpha i}{}^{2}+\frac{1}{2a_{i}{}^{2}}e_{\alpha i}{}^{2}\Theta_{i}\psi {\left(x_{i}\right)}^{T}\psi \left(x_{i}\right)+\frac{1}{2}{\bar{\lambda }}_{i}{}^{2}-\frac{1}{\sigma_{i}}{\tilde{\Theta }}_{i}{\dot{\hat{\Theta }}}_{i} \end{array}$$Since $\left|\frac{\lambda_{2}\left(t\right)m}{1+\lambda_{i}\delta }\right|\leq \frac{m}{1-\delta }$ and $\frac{w_{i}\left(t\right)}{1+\lambda_{i}\delta }\leq \frac{w_{i}\left(t\right)}{1+\delta }$, one has
(38)$$\begin{array}{ll} {\dot{V}}_{3} & \leq \Xi_{\Lambda i}\left(e_{\alpha i}+\alpha_{i}\right)+\left(1-q\left(e_{xi}\right)\right)[\frac{2L_{Li}{\dot{L}}_{Li}}{\pi }\tan\left(\frac{\pi e_{xi}{}^{2}}{2L_{Li}{}^{2}}\right)-\Lambda_{Li}\left(\left(\sum_{j=1}^{N}a_{ij}v_{j}+\sum_{r=1}^{M}b_{ir}v_{r}\right)+e_{xi}\frac{{\dot{L}}_{Li}}{L_{Li}}\right)]\\ & +q\left(e_{xi}\right)[\frac{2L_{Ui}{\dot{L}}_{Ui}}{\pi }\tan\left(\frac{\pi e_{xi}{}^{2}}{2L_{Ui}{}^{2}}\right)-\Lambda_{Ui}\left(\left(\sum_{j=1}^{N}a_{ij}v_{j}+\sum_{r=1}^{M}b_{ir}v_{r}\right)+e_{xi}\frac{{\dot{L}}_{Ui}}{L_{Ui}}\right)]\\ & +e_{\alpha i}(M_{i}^{-1}\left(-\frac{e_{\alpha i}u_{fi}{}^{2}}{\sqrt{e_{\alpha i}{}^{2}u_{fi}{}^{2}+\partial^{2}}}-\frac{e_{\alpha i}{\bar{m}}^{2}}{\sqrt{e_{\alpha i}{}^{2}{\bar{m}}^{2}+\partial^{2}}}+\left|\frac{\lambda_{2}\left(t\right)m}{1-\delta }\right|\right)\\ & -{\dot{\alpha }}_{i})+\frac{1}{2a_{i}{}^{2}}e_{\alpha i}{}^{2}\Theta_{i}\psi {\left(x_{i}\right)}^{T}\psi \left(x_{i}\right)+\frac{1}{2}a_{i}{}^{2}+\frac{1}{2}e_{\alpha i}{}^{2}+\frac{1}{2}{\bar{\lambda }}_{i}{}^{2}-\frac{1}{\sigma_{i}}{\tilde{\Theta }}_{i}{\dot{\hat{\Theta }}}_{i} \end{array}$$According to **Lemma 1**, we obtain
(39)$$\begin{array}{ll} {\dot{V}}_{3} & \leq \Xi_{\Lambda i}\left(e_{\alpha i}+\alpha_{i}\right)+\left(1-q\left(e_{xi}\right)\right)[\frac{2L_{Li}{\dot{L}}_{Li}}{\pi }\tan\left(\frac{\pi e_{xi}{}^{2}}{2L_{Li}{}^{2}}\right)-\Lambda_{Li}\left(\left(\sum_{j=1}^{N}a_{ij}v_{j}+\sum_{r=1}^{M}b_{ir}v_{r}\right)+e_{xi}\frac{{\dot{L}}_{Li}}{L_{Li}}\right)]\\ & +q\left(e_{xi}\right)[\frac{2L_{Ui}{\dot{L}}_{Ui}}{\pi }\tan\left(\frac{\pi e_{xi}{}^{2}}{2L_{Ui}{}^{2}}\right)-\Lambda_{Ui}\left(\left(\sum_{j=1}^{N}a_{ij}v_{j}+\sum_{r=1}^{M}b_{ir}v_{r}\right)+e_{xi}\frac{{\dot{L}}_{Ui}}{L_{Ui}}\right)]\\ & -M_{i}^{-1}\left|e_{\alpha i}u_{fi}\right|-e_{\alpha i}{\dot{\alpha }}_{i}+\frac{1}{2a_{i}{}^{2}}e_{\alpha i}{}^{2}\Theta_{i}\psi {\left(x_{i}\right)}^{T}\psi \left(x_{i}\right)+\frac{1}{2}a_{i}{}^{2}+\frac{1}{2}e_{\alpha i}{}^{2}+\frac{1}{2}{\bar{\lambda }}_{i}{}^{2}-\frac{1}{\sigma_{i}}{\tilde{\Theta }}_{i}{\dot{\hat{\Theta }}}_{i}+2M_{i}{}^{-1}\partial \end{array}$$Similar to (31)–(33), one has
(40)$${\dot{V}}_{3}\leq -\frac{\pi }{rT_{c}}\left(V_{3}{}^{1-\frac{r}{2}}+V_{3}{}^{1+\frac{r}{2}}\right)+C_{i}$$
where $C_{i}=\frac{1}{2}a_{i}{}^{2}+\frac{1}{2}{\bar{\lambda }}_{i}{}^{2}+\frac{\pi }{rT_{c}}\left(\frac{1}{2^{-\frac{r}{2}}}\Theta_{i}{}^{2-r}+\frac{1}{2^{\frac{r}{2}}}\Theta_{i}{}^{2+r}\right)+2M_{i}{}^{-1}\partial$.The proof is performed. □

## 4. Simulations

To verify the validity of the above results, a simulation is presented on multi-robot systems with eight single-link robots. The MRSs are composed of four follower robots and four leader robots. The directed topologies G are displayed in Figure 1, and the topologies switch sequentially every 10 s.

The dynamics of the i-th follower robot are expressed as
(41)$$\left\{ \begin{array}{l} {\dot{x}}_{i}=v_{i},\\ {\dot{v}}_{i}=J_{i}{}^{-1}(u_{i}(t)-B_{i}v_{i}-M_{i}gl_{i}\sin(x_{i})+\ell_{i}), \end{array}\right.$$
where $J_{i}$ is the total rotational inertia. $x_{i}$ and $v_{i}$ denote the angle and angular velocity of the i-th follower robot, respectively. *ℓ_i_* is the external disturbance. $u_{i}$ denotes the control input of the i-th follower robot. $M_{i}$ and $B_{i}$ denote the mass and damping coefficient, respectively. $l_{i}$ denotes the distance between the joint axis and the mass center.

The initial state of the followers is $v(0)={\left[0.9,\;0.8,\;0.6,\;1\right]}^{T},\;x(0)={\left[-1.4,0.5,1.9,\;-1.9\right]}^{T}$. The r-th leader robot’s predefined trajectory is $x_{r}=\sin(0.5t)$. The i-th follower robot model parameters are set as follows: $J_{i}=1$, $B_{i}=1$, $M_{i}=1$, g=10, $l_{i}=1$, $\ell_{i}=5\sin(t)$. The i-th follower robot controller $u_{i}$ is set as follows: r=0.4, $T_{c}=0.8$, $\sigma_{i}=500$, $a_{i}=1$, and $L_{Li}=10e^{-3t}+2,$ $L_{Ui}=8e^{-3t}+1$. Input saturations are defined as $u_{i}^{+}=100N\cdot m$, $u_{i}^{-}=-100N\cdot m$.

The tracking of the followers’ angle and angular velocity are shown in Figure 2 and Figure 3, respectively. The tracking errors all converge in the predefined time $2T_{c}=1.6\;s$. The tracking errors of the angle and angular velocity are represented in Figure 4 and Figure 5, and the tracking errors of the angle do not violate the predetermined constraints. The lower boundary is *–L_U1,_* and the upper boundary is *L_L1_*. The control inputs of the system and their partial enlargement are shown in Figure 6. It is clearly seen that the control inputs are smooth.

The uncertainties and external disturbances of the multi-robot system are approximated using a predefined-time fuzzy logic system. Figure 7, Figure 8, Figure 9 and Figure 10 show the approximation effect of the technique on the multi-robot systems’ external disturbances and uncertainties, respectively. Figure 11 shows the approximation error of the technique. From this, we know that the technique has a good approximation effect and that the approximation error is able to converge in a predefined time $2T_{c}=1.6\;s$.

To save communication bandwidth, the relative threshold event-triggered controller is used, and the parameters of the controller are set as follows: $\delta =0.05,\;\partial =2,\;\bar{m}=1.1,\;m=1$. After adding the relative threshold event-triggered controller, the angle consensus and angular velocity consensus of the i-th follower robot are shown in Figure 12 and Figure 13. From Figure 14 and Figure 15, all tracking errors, including angle errors and angular velocity errors, can converge to near zero in the predefined time of 1.6 s. As shown in Figure 16, the control inputs to the system appear in steps under the control of the event-triggered controller. The trigger times and the trigger intervals of each follower robot are depicted in Figure 17. It is calculated that each follower controller saves 85.6%, 85.5%, 85.1%, and 84.5% of the communication bandwidth.

To demonstrate the superiority of the proposed event-triggered controller, it is compared with the relative threshold event-triggered controller in [39]. The communication bandwidth savings of each follower controller are compared while accomplishing a similar angle-tracking effect. Figure 18a,b show the angle-tracking effect under two event-triggered controllers. After calculation in Table 1, with the controller in [39], the communication bandwidth is reduced by 68%, 70.7%, 66.8%, and 71.3%, respectively. A comparison of the specific trigger cases is shown in Figure 19a,b. Therefore, the proposed algorithm has better performance.

To demonstrate the superiority of the proposed predefined-time controller, it is compared with a fixed-time controller. Figure 20 and Figure 21 show the angle-tracking effect and the control input under the two controllers. Provided that the control inputs are of similar size, the follower’s angle keeps up with the leader faster under the action of a predefined-time controller. After calculation, we obtain the theoretical fixed-time bound as 20.2 s. The simulation results show that the convergence time of the whole system is approximately 2.5 s. The fixed-time upper bound and the actual convergence time indicate that the fixed-time convergence theory is very conservative in estimating the settling time bounds. The tracking errors all converge in the predefined time of 1.6 s. In contrast, the proposed predefined-time controller is more advantageous.

The IAE and ITAE of the angular and angular velocity errors are given under different controllers, and the comparison shows that the convergence error is smaller and the controllers are superior under the proposed predefined-time controller in Table 2, Table 3, Table 4 and Table 5.

## 5. Conclusions

This article proposes a predefined-time adaptive fuzzy cooperative tracking controller for multi-robot systems. The developed asymmetric tan-type BLF technique is applied to limit the error of the system, the proposed predefined-time convergent fuzzy logic systems can effectively handle the system uncertainties, input saturation, and external disturbances, and the designed relative threshold event-triggered controller largely reduces communication bandwidth. The predefined-time stability theorem is employed to demonstrate that the errors of consensus globally converge to near zero in predefined time. Some simulation examples on MRSs are given to illustrate the effectiveness of the above results. Transient behavior analysis and deep reinforcement learning will be considered in the future.

## Figures and Tables

**Figure 1 sensors-23-07950-f001:**
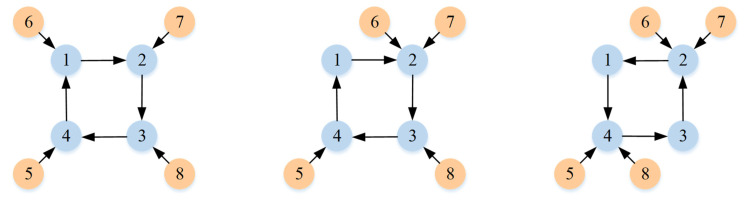
The directed topology.

**Figure 2 sensors-23-07950-f002:**
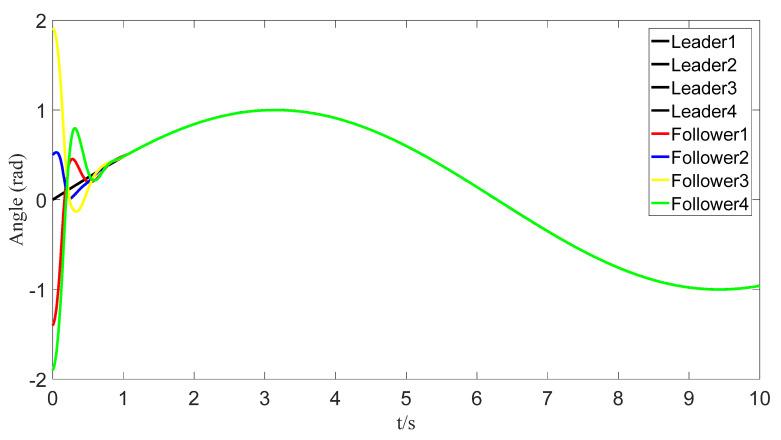
Angle-tracking effect graph.

**Figure 3 sensors-23-07950-f003:**
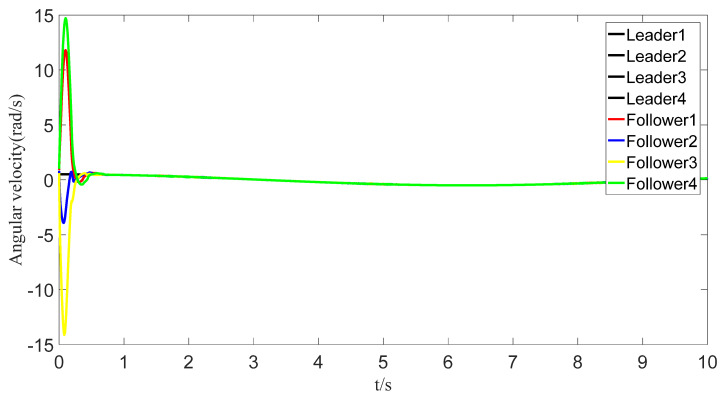
Angular velocity-tracking effect graph.

**Figure 4 sensors-23-07950-f004:**
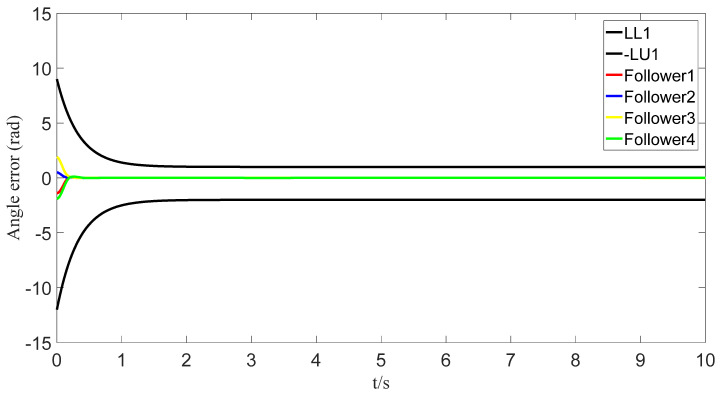
Angle-tracking error graph.

**Figure 5 sensors-23-07950-f005:**
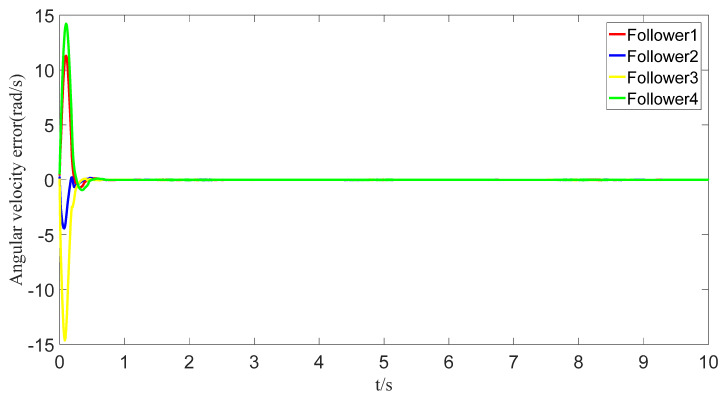
Angular velocity-tracking error graph.

**Figure 6 sensors-23-07950-f006:**
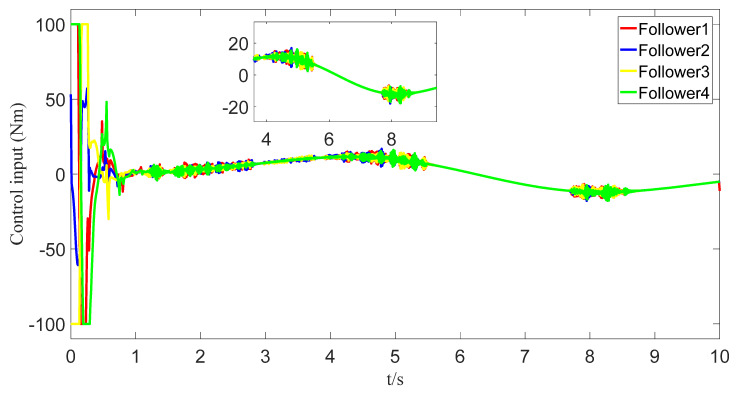
Control input graph.

**Figure 7 sensors-23-07950-f007:**
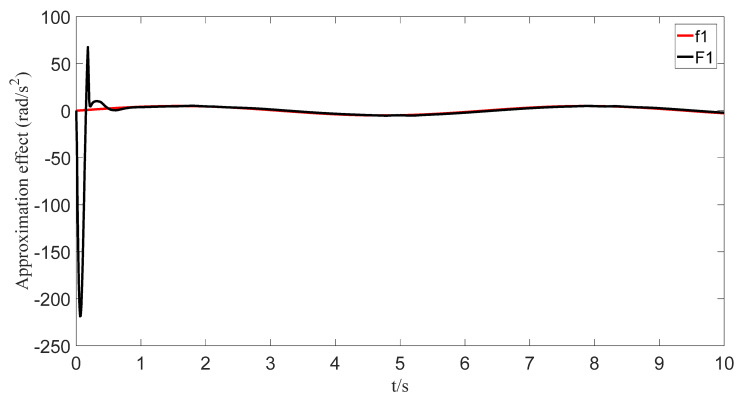
Fuzzy logic system approximation graph (f_1_).

**Figure 8 sensors-23-07950-f008:**
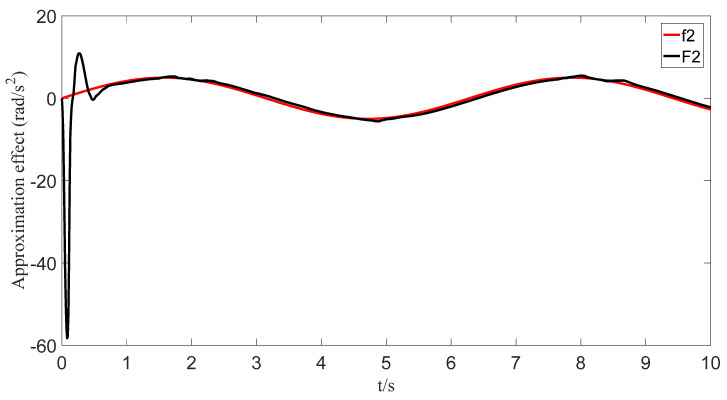
Fuzzy logic system approximation graph (f_2_).

**Figure 9 sensors-23-07950-f009:**
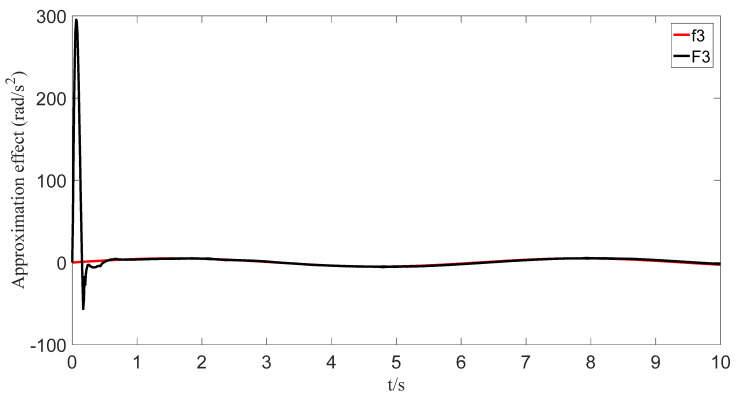
Fuzzy logic system approximation graph (f_3_).

**Figure 10 sensors-23-07950-f010:**
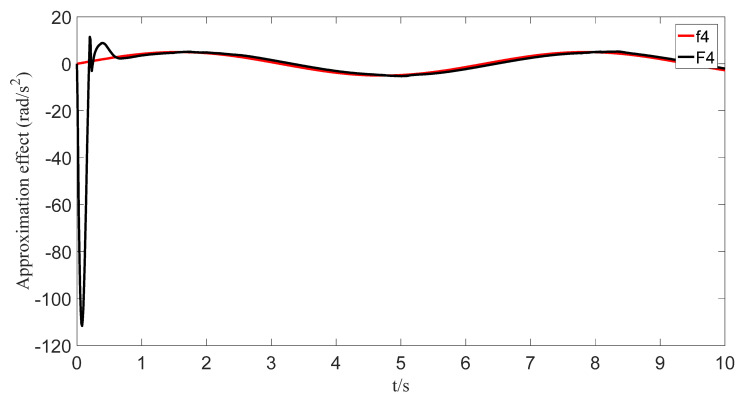
Fuzzy logic system approximation graph (f_4_).

**Figure 11 sensors-23-07950-f011:**
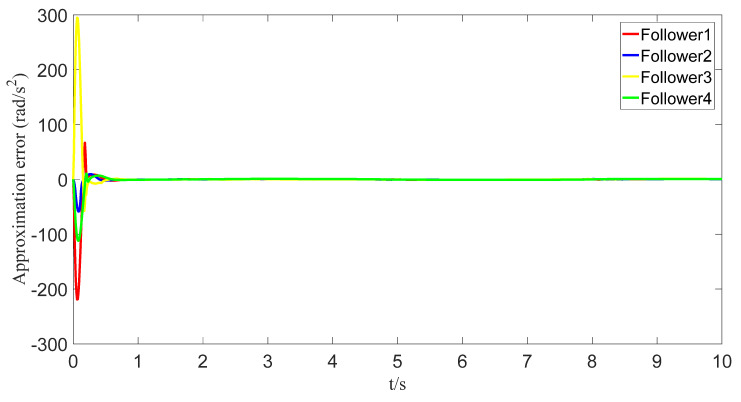
Fuzzy logic system approximation error graph.

**Figure 12 sensors-23-07950-f012:**
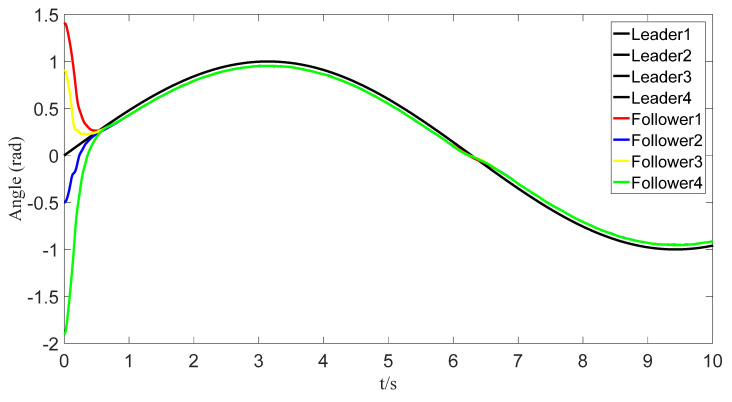
Tracking of angles with event triggering.

**Figure 13 sensors-23-07950-f013:**
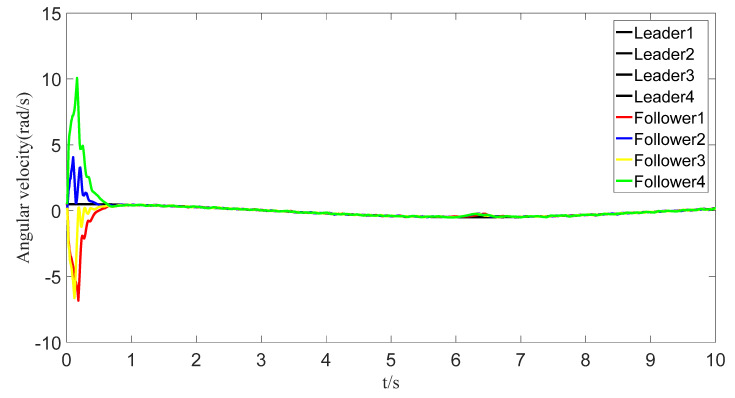
Tracking of angular velocities with event triggering.

**Figure 14 sensors-23-07950-f014:**
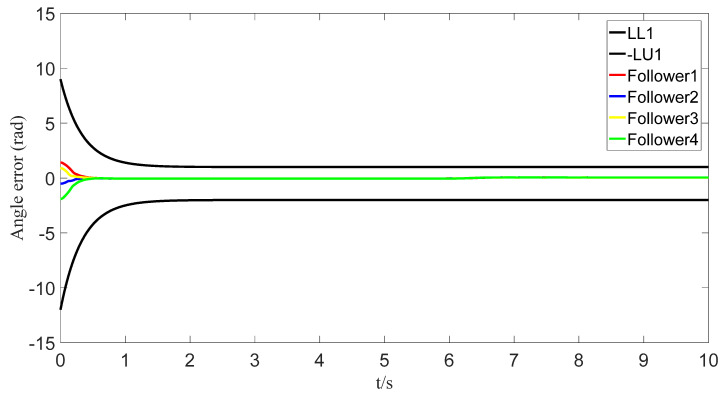
Error of angles with event triggering.

**Figure 15 sensors-23-07950-f015:**
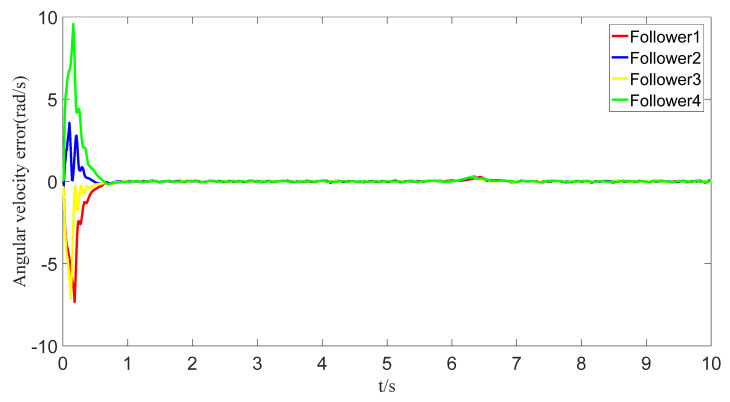
Error of angular velocities with event triggering.

**Figure 16 sensors-23-07950-f016:**
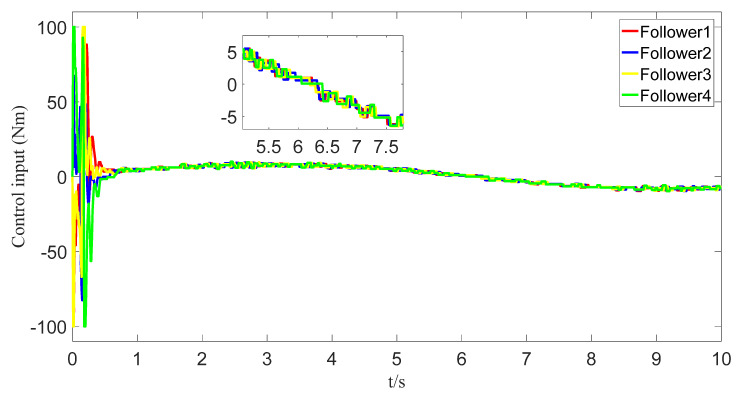
The control input with event triggering.

**Figure 17 sensors-23-07950-f017:**
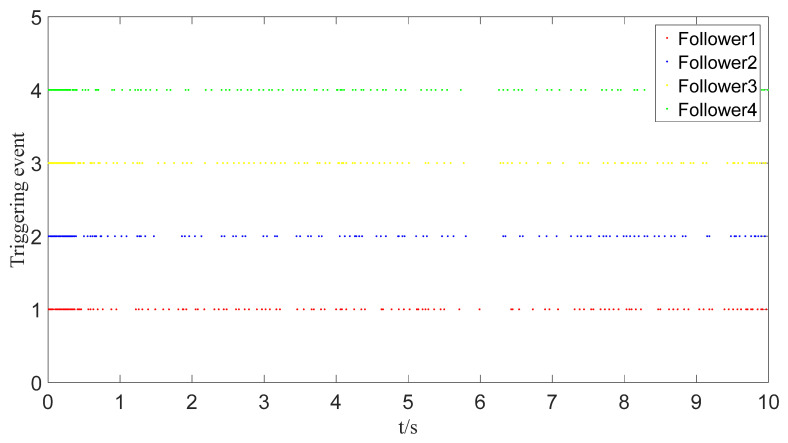
Triggering event graph.

**Figure 18 sensors-23-07950-f018:**
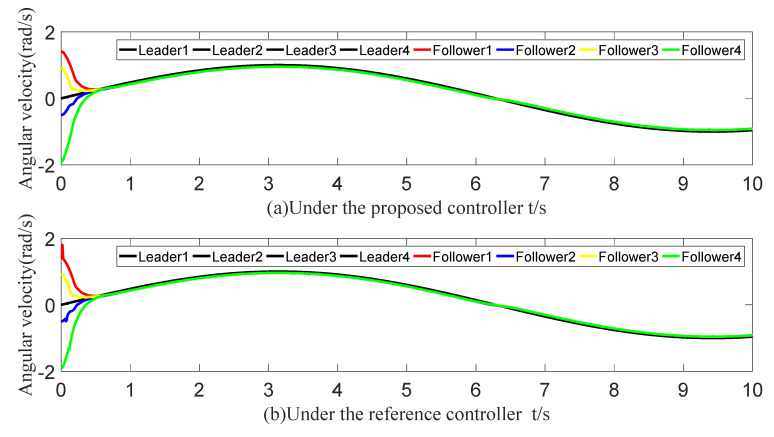
Comparison graph of angle tracking.

**Figure 19 sensors-23-07950-f019:**
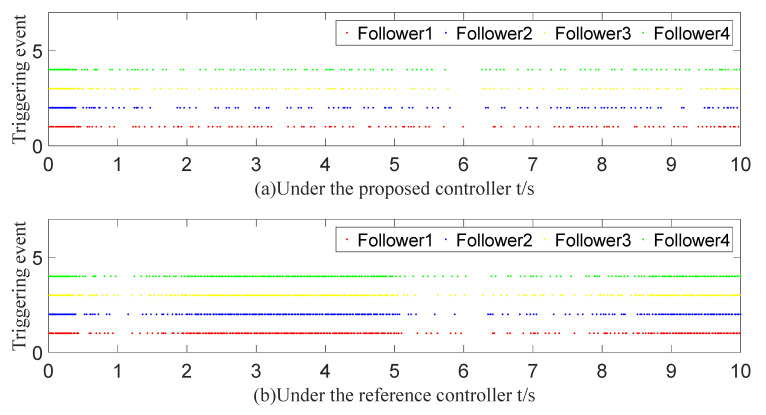
Comparison graph of triggering events.

**Figure 20 sensors-23-07950-f020:**
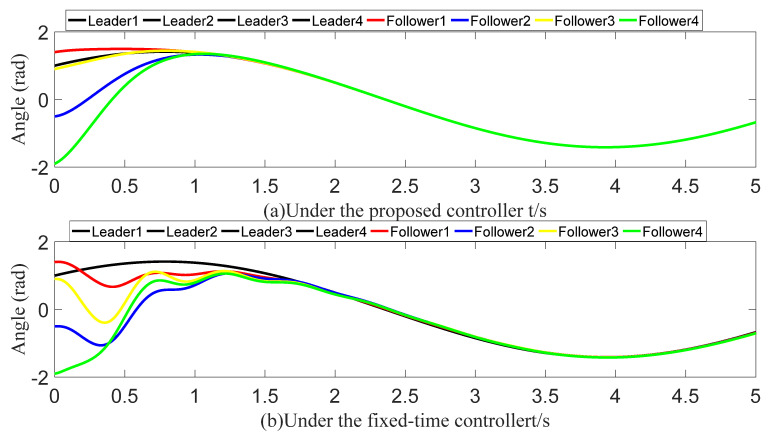
Angle-tracking effect graph (comparison).

**Figure 21 sensors-23-07950-f021:**
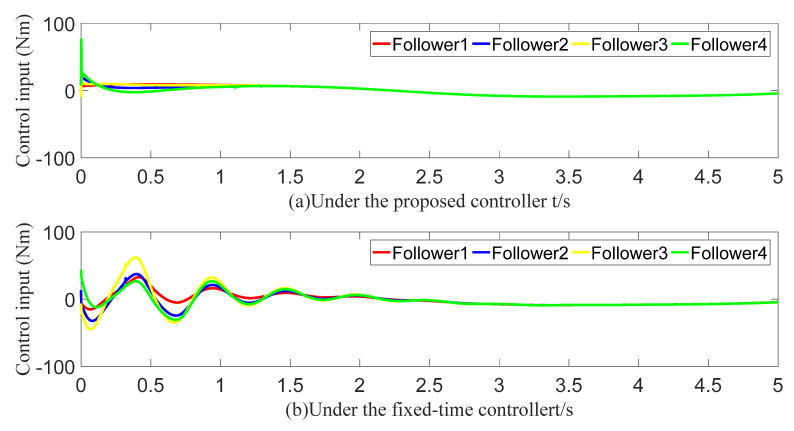
Control input graph (comparison).

**Table 1 sensors-23-07950-t001:** Comparison of bandwidth saving percentage.

*i*	Dynamic ETM	ETM in [39]
1	85.6%	68%
2	85.5%	70.7%
3	85.1%	66.8%
4	84.5%	71.3%

**Table 2 sensors-23-07950-t002:** IAE for angle error.

*i*	Predefined-Time Controller (rad)	Fixed-Time Controller (rad)
1	0.1466	0.6648
2	0.0491	1.9050
3	0.1897	1.2000
4	0.2125	2.0620

**Table 3 sensors-23-07950-t003:** IAE for angular velocity error.

*i*	Predefined-Time Controller (rad/s)	Fixed-Time Controller (rad/s)
1	1.6230	2.0540
2	0.5945	3.5950
3	1.9640	4.4800
4	2.2020	3.5730

**Table 4 sensors-23-07950-t004:** ITAE for angle error.

*i*	Predefined-Time Controller	Fixed-Time Controller
1	0.0250	1.3000
2	0.0184	2.1630
3	0.0305	1.9910
4	0.0363	1.9630

**Table 5 sensors-23-07950-t005:** ITAE for angular velocity error.

*i*	Predefined-Time Controller	Fixed-TIME Controller
1	0.4265	2.1810
2	0.2945	3.8510
3	0.4031	4.2800
4	0.4923	3.8650

## Data Availability

Not applicable.

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
