# Peer review of "Adaptive Fuzzy Event-Triggered Cooperative Control for Multi-Robot Systems: A Predefined-Time Strategy"

_sensors, 2023, doi:10.3390/s23187950_

Round 1
Reviewer 1 Report
Please refer to the uploaded attachment for specific comments

Please refer to the uploaded attachment for specific comments
Reviewer 2 Report
I have the following comments for this submission:
1. At the beginning, the control input is extremely high. For practical applications, control signals of such high amplitudes are not desirable. The authors should consider this point with utmost sincerity and provide a solution.
2. The authors should extensively compare their results with others, not just with one.
3. Compare results in terms of different error indices like IAE, ITAE, ISE, ITSE, etc.
4. Provide control signal energies with the proposed and existing control schemes.
5. What are the disadvantages of the proposed scheme?
6. Units of ylabels are missing.
7. For better readability, include lists of symbols and abbreviations.
8. In several places, legends cover the figures, which should be rectified. The authors may change the legends of a graph from vertical to horizontal.
9. References should have uniform formatting.
10. The paper contains several equations. To avoid any mistakes, recheck the equations minutely.
11. A thorough proofreading is required to rectify the typos.
12. The English and the overall outfit of the manuscript should be improved.
See the main report.
Reviewer 3 Report
This article studies the predefined-time adaptive fuzzy event-triggered cooperative control problem for multi-robot systems that takes into account external disturbances and model uncertainties. After my reviewing, the work is practical and interesting, but it can not be accepted in its current version and some following comments are required to increase the quality of the article.
1. The authors claim that a dynamic event-triggered mechanism is proposed to reduce the triggering events. According to the triggering conditions in (33)-(35), the triggering threshold parameters $\delta$ and m both are constant. Why the authors name the ETM as dynamic ETM? Please give some explanations.
2. The controlled error states converge to zero very fast. Is the system a stable one? The authors should provide the error states without control and show the effectiveness of the proposed control strategy.
3. The literature reviewing is insufficient. Some recent works about event-triggered control are required to be discussed to update the reviewing. For example, Sampled memory-event-triggered fuzzy load frequency control for wind power systems subject to outliers and transmission delays.
4. The figure qualities of some simulation results can be improved. For, example, the simulation time seems too long to see the state regulation process clearly.
5. A table should be given to show the specific number of triggering times for different followers under different ETMs.
Minor editing of English language required.
Round 2
Reviewer 1 Report
No further comments.
Author Response
Thank you very much.
Reviewer 2 Report
The revision is not at all satisfactory.
1. The initial control effort is still high. Revise the manuscript to resolve this issue.
2. Not just in terms of IAE, compare the results in terms of other error indices, as mentioned earlier.
3. What about control signal energies? (Note that the energy of a continuous-time signal is defined as the area under the squared magnitude of the considered signal.)
4. The paper still contains several typos. A thorough proofreading is mandatory.
5. I could not find any list of symbols/abbreviations in the revised manuscript. Include one list for symbols and another one for abbreviations.
6. In Reference [33], pp. is wrong. Vol and issue no. are also missing. Sometimes, journal names are given in full; sometimes, abbreviations are used for them. Reference [43] is incomplete. Have the authors really checked the manuscript before submission? The authors must understand that such mistakes create a very bad impression.
Concluding remark: The paper can not be accepted unless the manuscript is thoroughly revised. The authors must address all the issues satisfactorily before the reviewer informs the final decision.
See the main report.
Reviewer 3 Report
Accept.
Author Response
Thank you very much.